# Increase of Glycoalkaloid Content in Potato Tubers by Greening as a Method to Reduce the Spread of *Pectobacterium* and *Dickeya* spp. in Seed Production Systems

**DOI:** 10.3390/microorganisms11030605

**Published:** 2023-02-27

**Authors:** Dorota Sołtys-Kalina, Anna Grupa-Urbańska, Renata Lebecka, Maud Tallant, Isabelle Kellenberger, Brice Dupuis

**Affiliations:** 1Plant Breeding and Acclimatization Institute–National Research Institute, Platanowa 19, 05-831 Młochów, Poland; 2Agroscope, Plant Production Systems, Route de Duillier 50, 1260 Nyon, Switzerland; 3Agroscope, Plant Protection, Route de Duillier 50, 1260 Nyon, Switzerland

**Keywords:** blackleg, α-solanine, α-chaconine, “solanine”, greening, control

## Abstract

*Dickeya* and *Pectobacterium* species are the causal agents of blackleg and soft rot diseases. This article explores the possibility of using the glycoalkaloids (GAs) naturally produced by the potato tuber after the greening process as a blackleg control method. We first tested the effect of GAs extracted from four potato cultivars on the growth and viability of one *Dickeya* and one *Pectobacterium* strain in growth media. Then, four years of field experiments were performed in which the incidence of blackleg was assessed in plants grown from the seed tubers of cv. Agria that were subjected to various greening treatments. In the growth media, all GAs isolated from the four cultivars appeared to be bacteriostatic and bactericidal against both bacteria strains. The inhibitory effect varied among GAs from different cultivars. Except for a one-year field trial, the blackleg incidence was lower in plants grown from green seed tubers without the yield being affected. The blackleg control was marginal, probably due to the low production of GAs by the tubers of cv. Agria after greening. Based on our findings, seed tuber greening has a good potential for blackleg control after the identification of varieties that present optimal GA composition after greening.

## 1. Introduction

*Dickeya* and *Pectobacterium* species are the pathogenic agents of blackleg disease in the field and soft rot disease during potato storage [1]. Diseases caused by *Dickeya* and *Pectobacterium* species are responsible for losses of about EURO 46 M annually for the potato sector of the European Union [2]. There are no commercial products available to control the spread of these bacteria in seed potato production systems either in the field or during storage [3]. The main reason is that products cannot reach the bacteria already present in the plant’s vascular system [4]. Thus, there is a strong need to find efficient control methods against these diseases in order to reduce the losses they annually cause.

The potato naturally produces toxic compounds named glycoalkaloids (GAs) [5,6]. The main GAs in potato leaves, stems, and tubers are α-solanine and α-chaconine. Both molecules are triglycosides of solanidine, which is a steroidal alkaloid derived from cholesterol [6]. α-solanine and α-chaconine occur in about the same concentration in potato tissues [7]. Four other glycoalkaloids can be detected in potato tissues in a lower quantity: solasonine, solamargine, leptinine I, and leptine II [8]. A study performed in the 1950s demonstrated the toxicity of α-solanine on potato pathogens, such as *Fusarium* spp. and *Phytophthora infestans* [9]. In the same study, α-chaconine was shown to be approximately twice as toxic against *Fusarium coeruleum*, while solanidine was about tenfold less toxic than α-chaconine against the same pathogen. More recent studies confirmed the higher inhibitory effect of α-chaconine on other fungal pathogens of the potato, such as *Rhizoctonia solani* [10]. The same study revealed a synergistic effect of α-solanine and α-chaconine with a pronounced enhancement of the effect of any single glycoalkaloid when they are both present. In addition, Fewell and Roddick [10] showed that α-solanine and α-chaconine had reduced activity at lower pH. Many other studies showed the antifungal activities of GAs on potato pathogens [11,12,13,14,15,16,17,18,19].

Few studies describe a potential bactericidal or bacteriostatic effect of GAs on potato pathogens. Mckee’s [9] study, α-solanine was not toxic against young cultures of bacterial potato pathogens, such as *Pseudomonas* spp. and *Erwinia* spp. (the former name of *Dickeya* spp. and *Pectobacterium* spp.). In contrast, the antimicrobial effect of glycoalkaloids in peel extracts was observed for some bacteria: *Escherichia coli*, *Pseudomonas aeruginosa*, *Proteus* sp., *Klebsiella* sp., *Bacillus subtilis*, and *Staphylococcus aureus* [20]. Andrivon et al. [14] showed that potato cultivars with a high GA content were more resistant to soft rot infections. This correlation, however, did not prove that GAs were the metabolites responsible for the higher resistance of the varieties to soft rot. A recent study performed by Joshi et al. [21] showed that metabolites extracted from potato varieties of the *Solanum chacoense* species had an effect on *Pectobacterium brasiliense* by reducing the exoenzyme activity and virulence gene expression but did not reduce the multiplication of the bacteria. Most studies dedicated to the effect of GAs on microorganisms refer to plant pathogenic fungi (e.g., *Fusarium*) or human pathogenic bacteria [10,20]. Thus, the knowledge regarding the effect of GAs on *Pectobacterium* spp. and *Dickeya* spp. is far from complete. In addition, they are no studies exploring the role of potato greening on the control of blackleg disease in the field.

It is well-known that tubers exposed to light turn green due to the synthesis of chlorophyll [22]. It is also known that the α-solanine and α-chaconine content in green potatoes is, on average, 300% higher than in ungreened potatoes [22]. This GA content in green tubers may vary between varieties [23]. A higher GA content could have an effect on controlling *Pectobacterium* and *Dickeya* species in potatoes [14]. Therefore, it would be of interest to know whether a higher content of GAs in seed tubers would allow a reduction of the incidence of blackleg symptoms in the field.

In this research, we first tested the effect of GAs extracted from four potato cultivars, namely Irys, Mieszko, Owacja, and Tajfun, on the growth and viability of one *Dickeya* and one *Pectobacterium* strain in bacterial growth media. This laboratory experiment was followed by four years of field experiments in which the influence of the greening of potato seed tubers inoculated with a *Dickeya* strain on the development of blackleg symptoms was evaluated.

## 2. Materials and Methods

### 2.1. Isolation of Glycoalkaloids from Potato Leaves

Forty tubers of each potato cultivar Tajfun, Mieszko Irys, and Owacja were planted at the beginning of May in pots in a greenhouse. Twenty plants of the same size were placed in a climatic chamber under controlled conditions (14 h day period and 20 °C). Leaves from the middle and upper part of plants were collected at full anthesis, mixed, divided into 20 g portions, immediately frozen in liquid nitrogen, and stored at −80 °C.

The GAs were isolated according to [24] with some modifications. Leaf samples (20 g) were ground in liquid nitrogen, and GAs were extracted in 2% acetic acid for two hours on a magnetic stirrer. The extract was filtered through a filter paper (ø 15 µm) and then through a column with C18 to remove chlorophyll. The extract was adjusted to pH 10 with ammonium hydroxide. GAs were precipitated in a water bath at a temperature of 80 °C for 40 min and cooled at 4 °C overnight. The pellet of GAs was centrifuged for 20 min at 4500 rpm, dried at room temperature, and then dissolved in 75% ethanol to obtain a concentration of 50 mg mL^−1^. GA solutions were used for the analysis of in vitro growth and viability of bacteria.

### 2.2. In Vitro Growth of Bacteria

Two bacterial isolates were used, both highly aggressive to potato tubers: (i) *Pectobacterium brasiliense*, the strain Pcb3M16 [25] and (ii) *Dickeya solani*, the strain IFB0099 (syn. IPO2276, in the collection of the Plant Research International, Wageningen, The Netherlands) [26].

The bacterial suspensions were prepared from colonies cultured overnight on a Luria-Berthani (LB) agar medium at room temperature and adjusted in an LB medium to a concentration of 10^8^ colony-forming units (CFU) mL^−1^. The bacterial suspension was diluted 10 times in an LB broth medium. The prepared suspension was distributed to wells of microtiter plates with 200 μL in each well. For the bacterial suspension, GAs isolated from potato cultivars dissolved in ethanol were added at a concentration of 0.8 mg mL^−1^. To exclude the effect of the ethanol in which GAs were dissolved on bacteria growth, the same volume of 75% ethanol was added to control samples instead of GAs. The growth of bacteria was assayed in a microplate reader for absorbance detection (Tecan Infinite F50, Tecan Austria GMBH, Grödig/Salzburg, Austria) at a wavelength of 620 nm at starting point and after 24 h of incubation at 25 °C with shaking at 150 rpm min^−1^. A multiplication factor (MF) was calculated as a quotient of the OD_620_ after 24 h of incubation and the OD_620_ at the outset. The MF was used to assess the in vitro growth of two strains of bacteria from two different species in the LB medium with the addition of GAs originating from the leaves of the four different potato cultivars. The experiment was repeated twice. In each experiment, three biological and three technical replicates were performed for each GA originating from the four different potato cultivars and four control samples.

### 2.3. Viability of D. solani and P. brasiliense in Growth Media

The viability of *D. solani* and *P.brasiliense* cells in LB growth media with or without GAs was measured using CyStain™ BacCount Viable kit (Sysmex, Warsaw, Poland) according to the manufacturer’s protocol. The bacterial suspension was prepared as described above; the final concentration obtained was 10^8^ CFU mL^−1^. Bacterial suspension (200 μL) was distributed in 0.5 mL sample tubes. The GAs extracted from the cultivars Irys, Mieszko, Owacja, and Tajfun were added to the final concentration of 0.8 mg mL^−1^, while the control sample contained 75% ethanol (that was used as a solvent for GA) in the same volume as in the tested samples. The percentage of dead cells was measured 48 h after incubation of bacteria at 25 °C, on a CyFlow Space flow cytometer (Sysmex, Warsaw, Poland), equipped with a blue laser. Dot plots were analyzed using FlowMax software. The experiment was performed in six replications.

### 2.4. Recognition of Glycoalkaloids in Leaves of Potato Cultivars

The GA fraction was isolated from 300 mg of frozen leaves of cvs. Irys, Mieszko, Owacja, and Tajfun. Ground tissue was extracted in 3 mL of methanol: water solution (7:3), shaken on a laboratory shaker for 2 h, and filtered through filter paper. After extraction, samples were passed through sterilizing filters (0.2 μm, Nalgene™, Thermo Scientific™, Waltham, MA, USA). An equal amount of acetonitrile with 1% formic acid was added to 750 μL of a sample, and the sample was applied to the solid phase of QuEChERS (ECQUCHL12CT, United Chemical Technologies, Bristol, PA, USA) and shaken for 30 s on a vortex mixer. Then, the supernatant obtained after isolation of the GAs fraction was diluted 10-fold with methanol. HPLC-MS separation conditions and equipment for GA recognition were the same as described in Szajko et al. [27]. A semi-quantitative analysis of the GA content in potato leaves was performed at a peak area using the following scale: C = 0; 1 = 0 < C < 25,000; 2 = 25,000 < C < 50,000; 3 = 50,000 < C < 75,000; 4 = 75,000 < C < 100,000.

### 2.5. Field Trials with Artificial Greening

Seed tubers of the cultivar Agria were used for these trials. The tubers came from a commercial seed lot that did not show blackleg symptoms in the field the year before. In the first year of the experiment, 800 tubers of this lot were washed, dried at room temperature for 24 h, and then exposed to artificial light. After that, the tubers were dispatched in 8 white plastic crates and put at a distance of 50 cm from the neon lights (Aura Light; TS Spreme SE; Longlife D35; 35w/840/T/L/G13). This exposure to light was performed for 10 days in a chamber at 15 °C and 70% of relative humidity (RH). The absolute light density received by the seed tubers was about 28.4 µmol m^−2^s^−1^ (SpectraPen LM 510; from 400 to 700 nm). After greening, 400 of these tubers were inoculated by soaking with the strain *Dickeya dianthicola* 8823 at a concentration of 10^5^ CFU mL^−1^ using the same protocol as presented in [28]. This *Dickeya* strain was selected due to its previously proven high virulence in the field [29]. At the same time, 800 tubers that were not exposed to artificial light were also inoculated with the strain *D. dianthicola* 8823 using the same protocol. After inoculation, the tubers were dried at room temperature for one week. Then, 400 inoculated tubers not yet exposed to artificial light were exposed to light using the same protocol detailed above. Finally, 400 tubers remained ungreened and uninoculated. During the entire process, all the tubers of the trial were exposed to the same regime of temperature and relative humidity. This process resulted in 400 tubers for each of the following treatments: (i) tubers greened but not inoculated, (ii) tubers greened and then inoculated, (iii) tubers inoculated and then greened, (iv) tubers ungreened and uninoculated, and (v) tubers ungreened and inoculated. In the second year of the experiment, 500 tubers were used per treatment instead of 400.

About two weeks after the last greening process, all the tubers were planted in the field in Changins (Vaud, Switzerland) at an altitude of 430 m a.s.l. The experimental fields were managed following good agricultural practices. The tubers of the five above-mentioned treatments were planted in small plots of four rows, and each row consisted of 25 plants. Within each row, the plants were separated by a distance of 30 cm, and a distance of 75 cm was set between two rows. Each treatment plot was repeated four times in year one (2019) and five times in year two (2020). The plots were separated from each other from 1.5 to 3 m. The observations consisted of scoring the percentage of emergence, the number of stems on 10 plants, and the number of plants with blackleg symptoms five times at one-week intervals during the growing season. After harvesting, the tuber yield per lot was assessed for the following tuber sizes: <42.5 mm, 42.5–60 mm, 60–80 mm, and >80 mm.

### 2.6. Field Trials with Natural Greening

Again, seed tubers of the cultivar Agria were used for this second series of field trials. The tubers also came from a commercial seed lot that did not show blackleg symptoms in the field. In the year before the trial, 200 tubers were inoculated by soaking with the strain *Dickeya dianthicola* 8823 at a concentration of 10^5^ CFU mL^−1^ using the same protocol as presented in Gill et al. [28]. The 200 tubers were kept uninoculated. These tubers were planted in a field in Changins (Vaud, Switzerland) in rows of 100 plants. In each row, the plants were separated by 30 cm, and a distance of 75 cm was set between rows. Ten (trial 2) to 15 days (trial 1) before harvest, the plants of one inoculated row and one uninoculated row were dug up using a potato digger shaker and left on the ground for natural greening. During the growing season, the plants were treated weekly with 7 L of mineral oil in 300 L of the mixture (Zofal D; Syngenta) to control the spread of viruses. Otherwise, the experimental field was managed following good agricultural practices. After greening, the four rows were harvested separately, and tubers were stored at 4 °C and 70% RH until the planting of the trial the following year. For both years of the trial, seed tubers were produced with the following treatments: (i) inoculated and greened, (ii) inoculated and ungreened, (iii) uninoculated and ungreened, and (iv) uninoculated and greened. 

The following year, these seed tubers were planted in 5 replicates of plots of 100 plants. The experimental design was the same as the one used for the trials with artificial greening. The observations performed during the growing season and post-harvest were also the same. This trial was repeated for two consecutive years, namely 2021 and 2022.

### 2.7. Total Glycoalkaloid Assessment in Green Tubers of cv. Agria

Tubers of cv. Agria were used to assess the total GA (TGA) accumulation after exposure to artificial light and sunlight (natural light). For the artificial light, we used the same lighting system as that used for the field trials. The same four treatments were implemented for artificial light and sunlight, i.e., (i) 6 tubers were not exposed to light; (ii) 6 tubers were exposed to light for 5 days; (iii) 6 tubers were exposed to light for 10 days; and (iv) 6 tubers were exposed to light for 15 days. The average light intensity was 28.4 µmol m^−2^s^−1^ (SpectraPen LM 510, Photon Systems Instruments, Drásov, Czech Republic; from 400 to 700 nm). For the sunlight treatment, the light exposure was different from the field experiment. The tubers were placed in a greenhouse unit (Nyon, Switzerland) at 20 °C and 70% RH at the end of September 2022. The weather was generally cloudy during this period, and a once-off measure of light intensity was taken on the 29th of September at noon with an average value of 358.2 µmol m^−2^s^−1^ (SpectraPen LM 510; from 400 to 700 nm). After the exposure to light, all tubers were stored in the dark at 4 °C for 6 weeks before TGA analysis.

After exposure of the tubers to light, the TGA content was analyzed according to Andreu et al. [24]. The concentration of GAs was expressed as equivalent to α-solanine as mg kg^−1^ dry weight (DW). The experiment was performed in six repetitions.

In addition, green potato tubers of cv. Agria exposed for 15 days to natural and artificial lights were analyzed to quantify the abundance of the different GAs, namely leptinine I, leptinine II, solasonine, solamargine, α-solanine, and α-chaconine. The tubers were ground in liquid nitrogen, weighed into 0.3 g portions, and stored at −80 °C until use. Tuber samples were then extracted for 2 h in 3 mL of a mixture of methanol:water (7:3), then centrifuged for 10 min at 9500 rpm, and the supernatant was filtered through a sterilizing filter (0.2 μm, Nalgene™). To 750 μL of the sample, the same amount of acetonitrile acidified with 1% formic acid was added. The mixture was added into the QuEChERS (UTC) solid phase, shaken vigorously for 30 s, and centrifuged for 1 min at 8000 rpm. The supernatant was diluted 10-fold with methanol. HPLC–MS analysis was performed on a Dionex 3000 RS-HPLC equipped with a DGP-3600 pump, a WPS-3000 TLS TRS autosampler, a TCC-3000 RS column compartment (Dionex Corporation, Sunnyvale, CA, USA), and a Bruker micrOTOF-QII mass spectrometer (Bruker Daltonics, Billerica, MA, USA). The chromatographic column was a 50 × 3.1 (i.d.)-millimeter Thermo Scientific Hypersil GOLDc column with 1.9 μm particles (Part No. 25002-052130, Serial No. 0110796A6, Lot No. 10922). Results are expressed as peak area on the histogram and abbreviated as: C = 0; 1 = 0 < C < 25,000; 2 = 25,000 < C < 50,000; 3 = 50,000 < C < 75,000; 4 = 75,000 < C < 100,000.

### 2.8. Data Analysis

For all trials, the analysis of variance (ANOVA) was used for the determination of statistical differences between treatments. If the conditions for the application of the ANOVA were not fulfilled, an appropriate transformation of the variable was used (square root, arc sinus, or logarithmic transformation). When differences among treatments were characterized, a post-hoc mean comparison test was used (Dunnett or Duncan test).

For laboratory experiments, the graphical presentation of basic statistics was performed using Microsoft^®^ Excel^®^ for Microsoft 365 MSO.

For the field trials, the differences in the number of stems were based on the comparison of the average number of stems on 10 plants per plot. The disease assessment was based on a comparison of the percentage of blackleg disease at the last observation when the maximum incidence was reached. For yield assessment, we considered the commercial tuber size, meaning the tuber size between 42.5 and 80 mm.

All the statistical analyses were performed using the STATISTICA 13 software program.

## 3. Results

### 3.1. Growth Analysis of the Bacteria

All four GAs inhibited the growth of both tested bacterial strains (*P. brasiliense* Pcb3M16 and *D. solani* IFB0099), which was expressed by significantly lower MF values compared with the control, as indicated by Duncan’s post-hoc test (Figure 1). GAs from cv. Tajfun had the strongest inhibitory effect on the growth of both bacterial species; the MF value was 1.8 and 1.6 for *D. solani* and *P. brasiliense*, respectively (Figure 1). The weakest inhibition of bacterial growth was noted for GAs isolated from cv. Irys, MF = 2.9 for *D. solani* and MF = 4.4 for *P. brasiliense*. ANOVA revealed significant effects of GAs (*p* < 0.001) and bacteria strains (*p* < 0.01) as well as their interactions (*p* < 0.05) on MF values. 

### 3.2. Viability of the Bacteria

Both bacteria were variably affected by GAs isolated from different cultivars (Figure 2). The highest percentage of dead cells was observed for *P. brasiliense* after incubation with GAs from cv. Tajfun (6.0%), however the viability of *D. solani* was not affected (Figure 2). GAs isolated from cv. Irys significantly increased the percentage of dead cells of both bacteria, to 4.3% for *P. brasiliense* and 5.6% for *D. solani.* GAs from cv. Owacja generated a higher percentage of dead cells of *P. brasiliense* (4.1%) while GAs from cv. Mieszko induced cell death of *D. solani* (4.1%).

### 3.3. Identification of Glycoalkaloids Isolated from Four Potato Cultivars

In the leaves of tested potato cultivars, four different GAs were identified (Table 1). Three steroidal glycoalkaloids, solamargine, α-solanine, and α-chaconine, and one leptine glycoalkaloid: leptinine I, were identified. All cultivars contained α-solanine and α-chaconine; cv. Tajfun and Mieszko at similar proportions, 4:4 and 3:3, respectively. Cultivar Mieszko, additionally, had leptine I and cv. Irys, solamargine (Table 1).

### 3.4. Field Trials with Artificial Greening

No differences were detected in the number of stems between the different treatments (*p* > 0.05). The multiyear ANOVA revealed a “year” effect (*p* = 0.036), a “treatment” effect (*p* < 0.001), and an interaction between “year” and “treatment” (*p* = 0.018). Therefore, years were analyzed separately for a better understanding of this interaction. The year-by-year analysis showed a strong “treatment” effect for both years (*p* < 0.001). In 2019, it was noted that some blackleg symptoms were observed in the plots planted with tubers that were not greened nor inoculated (Figure 3A). Nevertheless, this percentage of blackleg was very low (1%). It suggests that the seed lot used for the trial was slightly infected or that some infection came from the neighboring plots. Nevertheless, this infection seems to have had no effect on the results, as both uninoculated treatments are statistically equal. The two greening treatments used in this trial, namely before and after inoculation, did not have a significant effect in preventing blackleg compared to the ungreened inoculated control (Figure 3). The results present a different trend for the two years. In 2019, there was less blackleg in the greening treatments (Figure 3A), while in 2020, less blackleg was observed in the treatment inoculated but without greening (Figure 3B). In addition, we noted that, for the year 2020, there was 10% more blackleg (19.4%) for the inoculation after greening treatment than in the ungreened inoculated control (9.5%).

The average commercial yield was higher in 2019 than in 2020 (*p* < 0.001). The greening had a positive effect on the yield in 2020 (*p* < 0.01). More potatoes of commercial tuber size were harvested in the plots where the seed potatoes were greened before planting, with an average of 14 tons more tubers per ha. This positive effect of greening on the yield was not observed in 2019. The inoculated plants had a lower yield with an average of 6 tons less of commercial yield/ha (*p* < 0.001). This drop in yield between the inoculated and the uninoculated plants was higher after greening, but as the average yield was higher for plants grown from green seed tubers, the commercial yield remained higher than for the ungreened treatment (see Figure 4).

### 3.5. Field Trials with Natural Greening

A significant difference between the number of stems between the two years of trials was detected (*p* < 0.001). In 2022, there was almost one more stem (0.9 stems more) per plant compared to the year 2021. The ANOVA also detected an interaction between the year and the greening effects (*p* < 0.01). In 2021, the number of stems was higher for the plants grown from green tubers, with, on average, 0.8 more stems, while no differences were detected among treatments in 2021.

Regarding the blackleg incidence, the multiyear ANOVA revealed a strong “year” effect (*p* < 0.001); therefore, each year was analyzed separately. The ANOVA showed differences between the treatments tested in 2021 (*p* < 0.001) but not in 2022 (*p* = 0.066) (Figure 5). In 2021 blackleg symptoms were observed in plants grown from the uninoculated and ungreened tubers (1.1%) and from the uninoculated-greened tubers (0.2%). (Figure 5A). These small infections can come from the seed lot chosen for this field trial or from neighboring plots. Nevertheless, this infection seems to have no effect on the results, as both uninoculated treatments are statistically similar. In 2021 the greening treatment did not allow a significant reduction in the expression of blackleg symptoms in the field (Figure 5A). The absence of significant differences between the treatments in 2022 is probably due to the low level of symptom expression for that year (Figure 5B). Despite the absence of a significant effect, fewer blackleg symptoms were observed from the tubers that underwent the greening process compared with plants from tubers that were kept in the soil until harvest. This trend was observed for both years of experiments (Figure 5).

The average commercial yield was higher in 2021 than in 2022 (*p* < 0.05), but no other yield effect was observed in this experiment.

### 3.6. Total Glycoalkaloid Assessment in Green Tubers of cv. Agria

TGA in tubers of cv. Agria significantly increased under both sunlight and artificial light, reaching a plateau on day 10 of exposure to the sunlight and a maximum on day 15 of exposure to artificial light (Figure 6). On day 15, tubers exposed to sunlight had an average of 1613 mg kg^−1^ DW. of TGA, and those exposed to artificial light had 2806 mg kg^−1^ DW.

Tubers of cv. Agria differed in GA composition and amounts, depending on the type of light they were exposed to (Table 2). Fifteen days after sunlight exposure, tubers accumulated solamargine, α-solanine, and α-chaconine in proportions 1:4:5. Artificial light induced the production of GA at a 1:5:5 proportion for solamargine, α-solanine, and α-chaconine. After exposure to artificial light, leptinine I, leptinine II, and solasonine were also detected, with a low and equal proportion of 1 (Table 2).

## 4. Discussion

Glycoalkaloids are the main bioactive compounds in potatoes. High GA content in potato tubers intended for consumption is not desired because of human toxicity. On the contrary, GA accumulation in above-ground organs is favorable since high GA content increases resistance to pathogens and herbivores [6]. Potato wild species, e.g., some *S. chacoense* accessions, are characterized by higher resistance to *Pectobacterium* spp. and *D. solani* than *S. tuberosum* accessions [30,31]. A high level of resistance to bacterial pathogens in plants is often correlated with high contents of secondary metabolites, such as phenolics, alkaloids, steroidal saponins, or phytoalexins [32]. Secondary metabolites alter virulence by regulating quorum sensing, affecting bacterial signaling and secretion systems, toxin production, motility, and extracellular enzymes, or act indirectly by destabilizing bacterial cell membranes and causing electrolyte leakage [21,33,34].

In this study, to determine whether GAs isolated from the leaves of potato cultivars affect *P. brasiliense* and *D. solani* virulence, MF and bacterial viability were analyzed and compared with GAs’ composition in each cultivar. Isolated GAs appeared to be bacteriostatic and bactericidal. They acted as bacterial multiplication inhibitors since neither bacteria were able to multiply effectively in vitro in the presence of GAs. The most pronounced lethal effects were observed for *P. brasiliense* after treatment with GAs isolated from cv. Tajfun. *Dickeya solani* viability was most affected by GAs isolated from cv. Irys. It was previously noted that GA abundance is correlated with bacteriostatic activity [21,35]. Tuber and stem extracts of *S. chacoense* line M6, highly resistant to *P. brasiliense,* contained more solamarine, α-solanine and β-chaconine than the susceptible *S. tuberosum* line DM1. The *S. chacoense* extracts had antimicrobial and antivirulence activities by reducing pectinase, cellulase, and protease activities of *P. brasiliense* in vitro [21]. This effect on bacterial communities was also observed in vivo. The content of α-solanine in potato tubers of cv. DaXiYang influenced the diversity and abundance of endophytic bacteria, as fewer operational taxonomic units (OTUs) were detected in the tubers with higher α-solanine contents [35]. In this study, both bacteria strains were treated with the same GA concentration. Thus, the glycoalkaloid composition was the main source of variation in bacterial response. We observed that an equal proportion of α-solanine and α-chaconine (4:4) was the most lethal combination for *P. brasiliense,* while solamarine, α-solanine and α-chaconine in proportions 1:2:3, were lethal for *D. solani*. Our findings are in line with other studies that describe various inhibitory activities of α-solanine, solasodine, and β-solamarine isolated from *S. dulcamara* against the bacteria *Staphylococcus aureus* [36]. β-solamarine significantly inhibited the growth of *S. aureus*, followed by α-solanine and solasodine. However, a lower concentration of α-solanine was required to induce the same bactericidal effect against *S. aureus* than solasonine and β-solamarine. Altogether, we have shown through these in vitro experiments that GAs isolated from potato leaves are compounds with antimicrobial and antivirulence activities. The biological effect they induce is mainly related to the type of glycoalkaloids present in the leaves.

After demonstrating that GA had a direct control effect on *Pectobacterium* and *Dickeya* in vitro, we wanted to verify whether the GA naturally produced by the potato tuber could result in a reduction of the incidence of blackleg disease in the field. As far as we know, this has never been tested before. For this experiment, we chose cv. Agria as it is known to be very susceptible to blackleg [37]. In order to increase the GA content in the seed tubers, they were exposed to artificial light (years 2019 and 2020) and sunlight (years 2021 and 2022). As mentioned above, GAs are toxic for human beings, which is the reason why greening must be strictly restricted to seed tubers that are not intended for consumption. The field experiments showed that the greening process did not have a negative effect on emergence or on the number of stems per plant. In 2021, more stems were observed on the plants grown from tubers naturally greened by the sun than on plants grown from ungreened tubers. This first observation is very important as each stem produces its own batch of tubers, and fewer stems could jeopardize the final yield [38]. This was confirmed after yield assessment, as there were no yield differences between plants produced from greened or ungreened seed tubers except in 2020 when a higher yield was obtained from potato plants grown from greened seed tubers. This implies that the greening treatment does not affect the productivity of the seed potato crop, whether natural or artificial greening.

Whatever the origin of the light (natural or artificial), fewer blackleg symptoms were observed for potatoes that were greened after infection or inoculation. This effect was observed in all years except for the second year of the experiment with artificial light (Figure 3A and Figure 5A,B). In the second year of the experiment, a significant increase in blackleg development was observed when inoculation occurred after greening with artificial light (Figure 3B). These contradictory results could be due to the weather conditions of the year. Nevertheless, for all years, the effect of blackleg reduction was not significant when the greening treatment was compared with the treatment “inoculated and ungreened” (*p* > 0.05 at Dunnett statistical test). From this field data, we would be tempted to conclude that greening does not have an effect in preventing blackleg development in the field. GA content in green potatoes varies considerably from one variety to the next [23,39,40,41,42]. The article of Griffiths, Dale, and Bain [42] explores the TGA increase of 20 different potato varieties after greening for 48 h (140 µmol m^−2^s^−1^ of artificial light). In this article, the lowest increase in TGA content is for cv. Aisla with 8 mg TGA/100 g DW, reaching 40 mg TGA/100 g DW, while the highest increase after greening is for cv. Arran Consul with 39 mg TGA/100 g DW up to 129 TGA/100 g DW. It means that cv. Arran Consul has 320% more TGA than cv. Aisla after greening. In our experiment, the increase in TGA of cv. Agria was dependent on the light source. The increase was faster for sunlight, with a 173% increase after 5 days, reaching a plateau after 10 days. There was a 313% increase after 10 days and a 314% increase after 15 days of exposure to sunlight. With artificial light, the increase was slower, with an 11% TGA increase after 5 days and a 131% increase after 10 days. The increase was spectacular after 15 days, with an 1176% increase in TGA content. Additionally, artificial light stimulated the accumulation of GAs other than α-solanine and α-chaconine, namely: leptinine I, leptinine II, and solasonine, and changed the proportions between them. Synthesis of selective GAs (α-solamarine and β-solamarine) was also noted in *S. phureja* tubers after exposure to artificial light [43]. These results demonstrate that cv. Agria produces GA when the tubers are exposed to light. From the blackleg data, it appears that the increase of GA content in the tubers of our experiment was not sufficient to provide efficient control of blackleg. This finding is supported by Kasnak and Artik [44], who showed that cv. Agria produces 12.5 mg of TGA/kg of fresh-weight of potato tubers after 14 days of exposure to fluorescent light, while cv. Bettina produces 37.4 mg/kg of TGA after the same period of exposure to fluorescent light. We can thus hypothesize that the use of greening of seed tubers to control blackleg in the field is only efficient with varieties producing a high amount of α-solanine and α-chaconine after greening.

In summary, this study suggests that potato greening could be used to control blackleg development in the field in seed potato production systems. Nevertheless, it still has to be demonstrated in the field with potato varieties producing more TGA after greening than cv. Agria. It is also necessary to determine the minimum TGA tuber production required to obtain significant control of the disease. Therefore, our field experiment should be repeated with varieties presenting a high accumulation of TGA after greening in the tuber periderm and flesh and the results of blackleg control compared with the results of varieties with a low accumulation of TGA after greening, such as cv. Agria. The confirmation of this finding would represent an important step forward in improving the quality of seed potato production in an ecological and sustainable manner. Our results indicate that the selection of varieties that possess the optimal GA composition and ability to accumulate a sufficient amount of GA after exposure to light is pivotal in reducing or eliminating the occurrence of *Dickeya* and *Pectobacterium* species in potato.

## Figures and Tables

**Figure 1 microorganisms-11-00605-f001:**
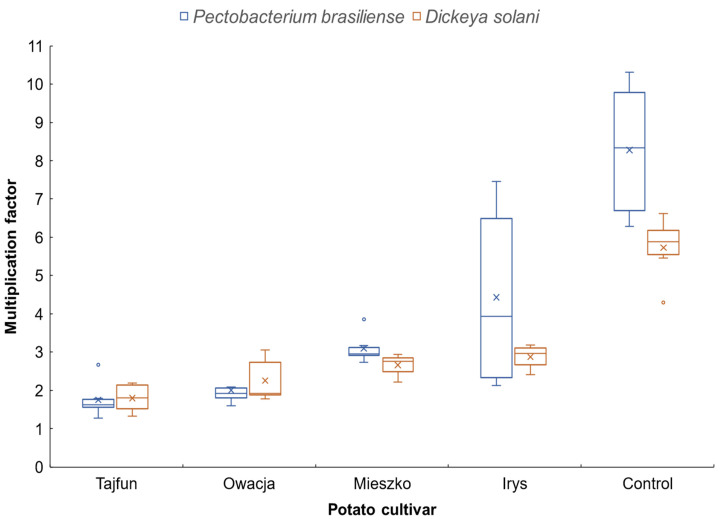
Average multiplication factor (MF) of *P. brasiliense* and *D. solani* 24 h after incubation in Luria Berthani medium (control) and Luria Berthani medium with glycoalkaloids isolated from leaves of four potato cultivars; ^x^ represents the mean of MF, error bars indicate the minimum and maximum values; the band inside the box represents median from nine replications in each of two independent experiments; the circles represent outliers.

**Figure 2 microorganisms-11-00605-f002:**
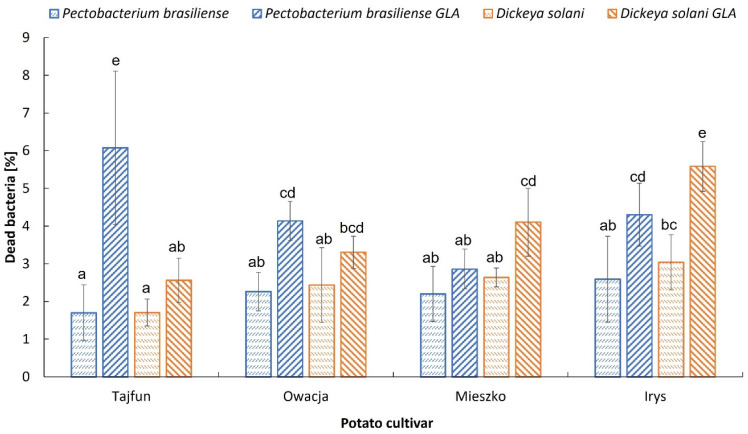
Percentage of dead cells of *P. brasiliense* and *D. solani* after 48 h growth in Luria Berthani medium (control) and Luria Berthani medium with glycoalkaloids isolated from leaves of four potato cultivars. Means marked with the same letter within strain do not differ significantly according to Duncan’s test at *p* = 0.05.

**Figure 3 microorganisms-11-00605-f003:**
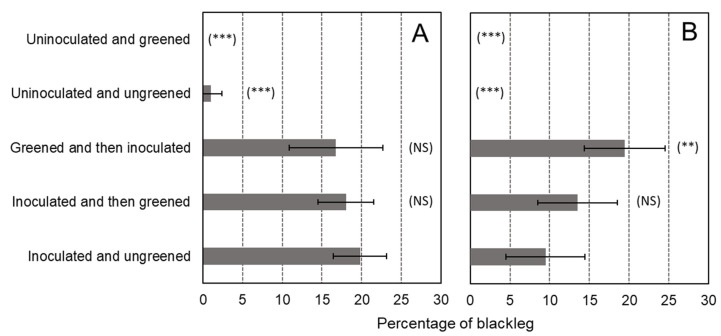
Percentage of blackleg in the trials of 2019 (**A**) and 2020 (**B**). The results of the mean comparison Dunnett test are presented in brackets: (NS) for “non-significant”; (**) for *p* < 0.01; (***) *p* < 0.001. Each treatment is compared with the inoculated and ungreened treatments. The error bars represent the standard deviations.

**Figure 4 microorganisms-11-00605-f004:**
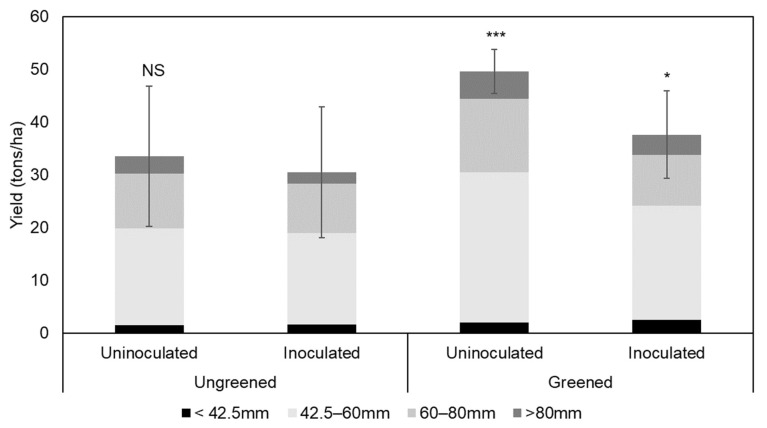
Average potato yield observed cumulatively for the years 2019 and 2020. The results of the mean comparison Dunnett test for the commercial yield (42.5 to 80 mm) are presented in brackets: (NS) for “non-significant”; (*) for *p* < 0.05; (***) *p* < 0.001. Each treatment is compared with the inoculated and ungreened treatments. The error bars represent the standard deviations of the commercial yield.

**Figure 5 microorganisms-11-00605-f005:**
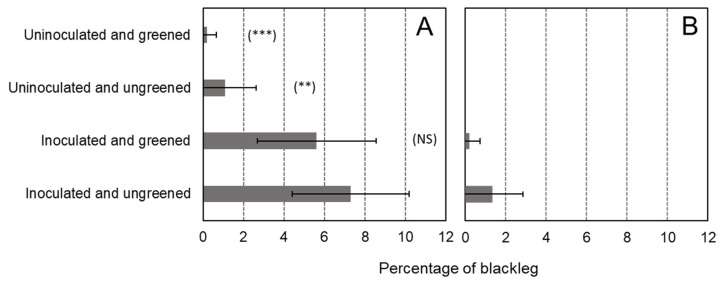
Percentage of blackleg in the trial of 2021 (**A**) and 2022 (**B**). The results of the mean comparison Dunnett test are presented in brackets: (NS) for “non-significant”; (**) for *p* < 0.01; (***) *p* < 0.001. Each treatment is compared with the inoculated and ungreened treatments. The error bars represent the standard deviations.

**Figure 6 microorganisms-11-00605-f006:**
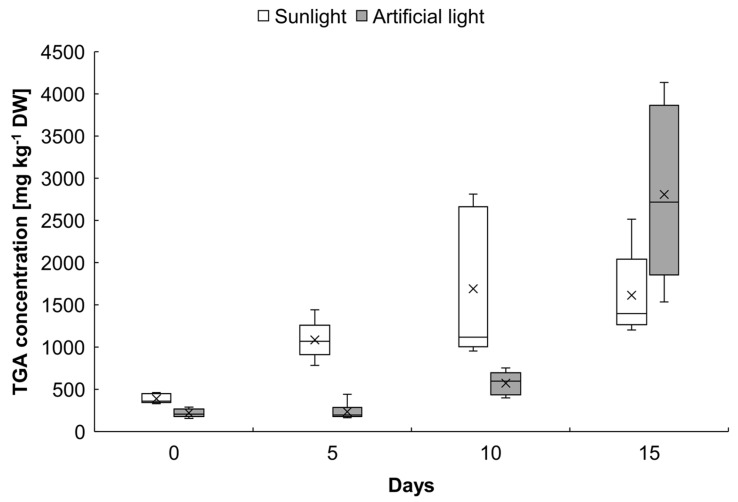
Total glycoalkaloid (TGA) concentration in tubers of cv. Agria in day 0 (ungreened tubers) and 5, 10, and 15 days after exposure to sunlight and artificial light. ^x^ represents the mean, error bars indicate the minimum and maximum values; the band inside the box represents median from six replications.

**Table 1 microorganisms-11-00605-t001:** Glycoalkaloids identified in potato cultivars and their amounts ^1^ in potato leaves.

Cultivar	Glycoalkaloid			
	Leptinine I	Solamargine	α-Solanine	α-Chaconine
Tajfun	0	0	4	4
Owacja	0	0	3	4
Mieszko	1	0	3	3
Irys	0	1	2	3

^1^ peak area on HPLC-MS histograms: C = 0; 1 = 0 < C < 25,000; 2 = 25,000 < C < 50,000; 3 = 50,000 < C < 75,000; 4 = 75,000 < C < 100,000.

**Table 2 microorganisms-11-00605-t002:** Glycoalklaoids detected in cv. Agria 15 days after exposure to sunlight or artificial light and their amount ^1^ in potato flesh.

Source of Light		Glycoalkaloid				
	Leptinine I	Leptinine II	Solasonine	Solamargine	α-Solanine	α-Chaconine
Sunlight	0	0	0	1	4	5
Artificial light	1	1	1	1	5	5

^1^ peak area on HPLC-MS histograms: C = 0; 1 = 0 < C < 25,000; 2 = 25,000 < C < 50,000; 3 = 50,000 < C < 75,000; 4 = 75,000 < C < 100,000.

## Data Availability

The data presented in this study are available on request from the corresponding author. The data are not publicly available due to privacy reasons imposed by the data policies of our respective Research Institutes.

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
