# Peer review of "Increase of Glycoalkaloid Content in Potato Tubers by Greening as a Method to Reduce the Spread of Pectobacterium and Dickeya spp. in Seed Production Systems"

_microorganisms, 2023, doi:10.3390/microorganisms11030605_

Round 1

Reviewer 1 Report

The manuscript titled ” Increase of glycoalkaloid content in greened seed potatoes as a method to reduce the spread of Pectobacterium and Dickeya spp. in seed production systems” represents an interesting study devoted to exploring the possibility of using the glycoalkaloids (GAs) naturally produced by the potato tuber after the greening process as a blackleg control method.

The manuscript is well-written and the results obtained support the conclusions raised by the Authors. However, there are some minor changes to be addressed:

Line 25: Please write keywords

Line 44: If you refer to several studies, please add more references in the bracket or replace More recent studies“ with More recent study“

Line 46-47: The same study revealed a synergistic effect of α-solanine and α-chaconine with a pronounced enhancement of the effect of any single glycoalkaloid, especially α-chaconine. “

The sentence is not clear. Please rephrase.

Lines 56, 153, 155, and 215: Missing numbers for references Al Kabee (2019), Gill et al. (2014), de Werra, (2021), and Andreu et al. (2001). Please align the rest of the reference numbers in the manuscript in accordance with this correction. The authors should also carefully check the whole text for such layout mistakes after the manuscript's acceptance.

Lines 56-57: Andrivon, Corbière, Lucas, Pasco, Gravoueille, Pellé, Dantec and Ellissèche [14]“. Instead of listing of all authors write Andrivon et al. [14]

Line 79: What is the reason for using only the Dickeya strain and not the Pectobacterium strain for field experiments as well? In vitro experiments were performed with strains of both bacterial species. Also, in in vitro experiments you used D. solani, not D. dianthicola. Can you explain the reason why field experiments were performed with the D. dianthicola strain?

Lines 98-100: Pectobacterium brasiliense, the isolate Pcb3M16, identified based on partial sequences of three conserved housekeeping genes, dnaX (DNA polymerase III subunit tau), icdA (isocitrate 100 dehydrogenase) and mDH2 (malate dehydrogenase) [25]“.
Was the used strain identified in the mentioned study conducted by Lebecka and Michalak (2020)? If so, the text regarding the identification of P. brasiliense Pcb3M16 can be deleted since all the needed details can be found in the cited work. If you performed PCR and sequencing for the purpose of this study, please add text explaining the methodology.

Line 122: Please replace “200 μL of the suspension” with “Bacterial suspension (200 μL)”

Lines 125-126: Please replace “with the same volume as in the tested samples“ with in the same volume as for the tested samples“

The authors should decide whether to use the term isolate(s) or strain(s) when referring to P. brasiliense Pcb3M16 and D. solani IFB0099 throughout the whole manuscript.

Lines 146 and 177: Replace ’variety’ with ’cultivar’ to standardize the terminology throughout the text

Lines 198 and 244: Subchapter 2.8. should be 2.7 and 2.9. should be 2.8.

Lines 201-203 and 207-209: “Four light treatments were implemented: (i) 5 tubers were not exposed to light, (ii) 5 tubers were exposed to neon light for 5 days; (iii) 5 tubers were exposed to neon light for 10 days; and (iv) 5 tubers were exposed to  neon light for 15 days.“
Instead of repeating the paragraph two times, for artificial light and sunlight, write: „The same four treatments were implemented for artificial light and sunlight, i.e. (i) 5 tubers were not exposed to light...“

Line 214: After exposure of the tubers to light, the TGA content was analyzed according to Andreu et al. (2001).
If the method was used without changes, there is no need to describe it in Lines 214-226.

Line 155: Replace the sentence “This Dickeya strain is known to be very aggressive in the field (de Werra, 2021).“ with „This Dickeya strain was selected due to its previously proven high virulence in the field (de Werra, 2021).“ and move it right after the sentence mentioning the strain for the first time.

Lines 261-263: The MF was used to assess the in vitro growth of two isolates of bacteria from two different species in the LB medium with the addition of GAs originating from leaves of the four different potato cultivars.“
Move the sentence to the section Material and methods and also explain the abbreviation upon the first use.

Lines 263-265: Replace All four GAs inhibited the growth of bacteria, which was expressed by a significantly lower MF value as indicated by Duncan's post hoc test (Figure 1.).“ with „All four GAs inhibited the growth of both tested bacterial strains (P. brasiliense Pcb3M16 and D. solani IFB0099), which was expressed by significantly lower MF values compared to the control, as indicated by Duncan's post hoc test (Figure 1.).“

Line 266: Replace bacteria species“ with bacterial species“

Line 286: Make P. brasiliense and D. solani italic in the sentance Percentage of dead cells of P. brasiliense and D. solani...“

Lines 326-327: Replace on average “ with „an average“

Line 332: Average potato yield for the years 2019 and 2020“ with Average potato yield observed cumulatively for the years 2019 and 2020“.

Line 368: You mentioned the abbreviation DW here for the first time, but you mentioned dry weight in line 225 for the first time. Please add the abbreviation DW in line 225.

Line 371: Replace “…..5, 10 and 15 days after exposure to sunlight or artificial light.“ with “…5, 10 and 15 days after exposure to sunlight and artificial light.“

Author Response

Dear reviewer,

Many thanks for your precious comments and for the many useful suggestions for text improvement. We have provided answers to all of them. The answers are provided in blue in the attached file.

The authors.

Reviewer 2 Report

On line 39, the authors state: The main GAs in potato leaves, stems and tubers are α-solanine and α-chakonine. Other GAs (Solamargine, Leptinine I) are also evaluated in the results. Can the authors add a short piece of information about minor potato alkaloids to the introduction.

Author Response

Dear reviewer,

Many thanks for your precious comment. The answer is provided in blue in the attached file.

The authors.

Reviewer 3 Report

The research is focused on the actual problem of preventing the losses in potato crops from bacterioses. In general, the purpose of the study is clearly formulated, and the literature is analyzed quite fully. I have some remarks.

1.On the basis of the results, the authors concluded that glycoalkaloids have a bactericidal effect on black leg pathogens. However, the percentage of dead pathogen cells in the presence of GA was only 1.5–3 times higher than the control values. For a valid conclusion, it is probably necessary to use higher concentrations of GA or potato varieties with a high content of GA.
2. An increase in the number of stems by 0.8–0.9 after treatment does not look very convincing, especially since these results have not been repeated over the years.
3. The contradictory results from field trials could be due to differences in weather conditions in different years, which must be taken into account when analyzing the data.
4. On page 6 L 270, the authors referred to Supplementary Table 1. Where is the Supplementary?
5. L 179 the authors wrote: "The year before the trial, 200 tubers were inoculated by soaking with the strain Dickeya dianthicola 8823..." Is this really true? How were tubers inoculated with a highly aggressive strain of Dickeya dianthicola stored?
How many tubers rotted during storage?

6. Check references, including italics for bacterial species

Author Response

Dear reviewer,

Many thanks for your precious comments. The answers are provided in blue in the attached file.

The authors
